# Vegetative Endotherapy—Advances, Perspectives, and Challenges

Jordana Alves Ferreira [1,2,*] , Llorenç Baronat Esparraguera [3], Sonia Claudia Nascimento Queiroz [1] and Carla Beatriz Grespan Bottoli [2]

[1] Laboratório de Resíduos e Contaminantes, Empresa Brasileira de Pesquisa Agropecuária—Embrapa Meio Ambiente, Rodovia SP 340, km 127.5, Jaguariúna 13918-110, Brazil; sonia.queiroz@embrapa.br
[2] Institute of Chemistry, University of Campinas, Campinas 13083-970, Brazil; carlab@unicamp.br
[3] ENDOterapia Vegetal, Calle Del Ripolles, 23, Castello Empuries, 17486 Gerona, Spain; llorens@endoterapiavegetal.com
[*] Correspondence: jordanaalvesferreira@gmail.com

**Abstract:** Vegetative endotherapy has shown satisfactory results in replacing conventional techniques for crop production material applications, such as spraying or via soil, in trees of perennial species. This review presents an overview of concepts and classifications for existing injection methods and covers applications from articles published in the last eighteen years on endotherapeutic techniques. An Excel interactive dashboard with data segmentation and filters to facilitate understanding of the data is provided. The indicators more relevant to researchers and producers, including the tree species evaluated, which were deciduous trees (24%), conifers (11%), ornamental (11%), and fruit trees (54%), are outlined. The most used products were insecticides, fungicides, and antibiotics, which are discussed. Pressurized and nonpressurized technologies were evaluated based on trunk opening, interface, and injection methods. And finally, an approach to good practices in precision agriculture is also discussed.

**Keywords:** trunk injection; tree injection; microinjection; trunk infusion; pressurized injection; systemic products; delivery products; nature-based solutions (NBS)

## 1. Introduction

Endotherapy comes from the Greek word "*endon*," which means endogenous, from the inside to the outside; while "*therapeia*" is the act of healing or restoring. Vegetative endotherapy, or just 'endotherapy' is the application of products within the trunk to deliver these products directly to the vascular system while water and nutrients are circulating during the photosynthetic cycle. Modern approaches to endotherapeutic treatment can increase both immunity and protection in plants by increasing resistance to harmful environmental conditions and mitigating the effect of pathogens such as fungi, bacteria, nematodes, insects, or viruses. Endotherapy is a treatment usually adopted in perennial crops, such as woody plants or palm plants, because, in most cases, these plants have the physical structure necessary for application in the adult/reproductive stage. This is in contrast to annual crops, which generally have life cycles and sizes that do not allow for this type of treatment, which can be expensive and labor-intensive. Endotherapy can be used in angiosperms such as deciduous, ornamental, and fruit trees and gymnosperms such as conifers. It is an alternative method to apply pesticides, nutrients, fertilizers, and phytohormones [1–7]. This phytosanitary treatment can replace conventional techniques such as spraying, fumigation, and/or soil drenches [8]. Critical factors for efficient endotherapeutic protocols include diagnosing the problem, such as determining if the issue is biotic or abiotic; understanding tree morphology and physiology; formulation, dosage, and intervals; injection method; seasons suitable for application and environmental conditions; and potential risk to non-target organisms [4,8–11]. Endotherapy is commonly known as

trunk injection or tree injection [6,12,13], pressurized treatments [14,15], therapeutic control [16], microinjection [17], macroinfusion [18], chemotherapy [19,20], and gravitational infusion [21–23].

The concept of endotherapy is not new: Leonardo Da Vinci (1452–1519) was one of the first to describe details of the systemic action of an arsenic solution by injecting it into the trunks of apple trees to contaminate their fruit [6,7,13,24,25]. Endotherapy can be performed using historical methods, such as a simple hand drill, syringe, and modern commercial equipment that assists in introducing products. In all cases, opening a port (hole) is necessary to introduce the products, which, after application, will translocate throughout the vascular system.

The injection method is vital as it influences the two functional processes that occur after a solution is injected into the trunk: (1) vertical uptake and (2) radial diffusion. Both are influenced by the injection method being used. Several studies have been conducted to evaluate new technologies and to improve existing techniques [4,26–28]. Some terms used to define the transport of products applied to the trunks are delivery [7,29] and translocation in the xylem [30]. In this review, it will be used the term translocation.

Endotherapy offers advantages over conventional treatments: it can reduce environmental contamination (dispersion or leaching), minimize health risks to humans and auxiliary fauna, be safer for the applicator, and deliver to the desired target with less loss [1,13,31]. No equipment is needed to reach the leaves that, depending on the crop, can be more than 10 m high above the ground. It allows localized treatment in situations of lethal and persistent attacks by arthropod pests and pathogens; is more persistent when leaf and/or soil treatments are complex or not effective; prevents the applied product from being washed away by rainwater or degraded by the sun (UV rays); has low water consumption [1,10,13,32,33].

The limitations of endotherapy are related to the need for more knowledge about the duration of the application and the lack of formulations. Furthermore, it requires individual treatment of every plant by perforation of the plant's trunk and the need to create a seal to prevent decomposition and infection by various pathogens. Information is also needed on endotherapy to avoid lesions in vascular tissue, especially in dicotyledons, that can lead to embolism, phytotoxicity, and even necrosis [13,24,25,31,32,34–36]. As vegetative endotherapy is a treatment, residues and ecotoxicological analysis must be carried out as in conventional methods, validating the use of the product in the system and evaluating pre-harvest intervals for fruits.

According to the document released by the International Union for Conservation of Nature (IUCN) [37], the definition of nature-based solutions (NBS) is:

"NBS are defined as actions to protect, sustainably manage, and restore natural or modified ecosystems, that address societal challenges effectively and adaptively, simultaneously providing human well-being and biodiversity benefits."

As the NBS has eight well-established principles, two of these could qualify endotherapy with ideas that would support the following [37]:

"Can be implemented alone or in an integrated manner with other solutions to societal challenges (e.g., technological and engineering solutions);

Are an integral part of the overall design of policies, and measures or actions, to address a specific challenge".

"NBS offer a group of solutions, among numerous others, to solve global societal challenges; NBS can complement, and be implemented alongside, other types of interventions".

Endotherapy could contribute to NBS for plants and improve dialogue to offer solutions that reduce environmental degradation and provide means to minimize harmful waste contamination. Climate change threats to plant health propose that endotherapy could be a sustainable alternative that has favored the preservation/maintenance of forest, arboreal, and urban biodiversity and agriculture of perennial plants in the control/management of species whose conventional treatments are ineffective. Implementing endotherapeutic equipment development practices associated with formulations using natural products,

biopesticides, essential oils, and plant volatiles are sustainable solutions. These results help the transformation process and improve food security/safety and pesticide reduction for human, animal, plant, and environmental health [37–39]. If endotherapy is considered a strand of the NBS, it could gain strength and attention from researchers and private and environmental institutions to contribute to sustainable agricultural practices.

For this reason, endotherapy needs precise definitions to avoid confusion regarding its concepts, which can prevent its advancement and acceptance. The operability of vegetative endotherapy lacks an objective definition and a set of principles so that it can be reproduced and executed accurately. In this context, the structure of this chapter includes three main components: (1) the classification of endotherapeutic systems; (2) the functioning of the technique based on the physiology and morphology of the plants; (3) the evaluation of publications on endotherapy in the last 18 years. The first version containing this technique definition framework was completed in 2020 during the pandemic and updated data for this publication in 2023.

*Classification of Endotherapeutic Systems*

Table 1 describes the classification of endotherapeutic systems based on (i) *trunk opening*, (ii) *interface method*; (iii) *injection method*, and (iv) *injection volume*.

(i)　*Trunk opening*: When the trunk is opened by drilling, the contact surface increases through the exposure of the conductive sap vessels, allowing the introduction of the applied products. Blades are drill-free ports, and perforations can be round-shaped, with a screw thread or lenticular shape. The term blade was considered in this article, as proposed by Montecchio [13], as a metal instrument designed to cut and/or pierce with impact and not with rotation as with drills. In this case, the blades remain in the trunk until the end of the application [1,13,32,40–42].

(ii)　*Interface methods*: Some technologies use removable injectors that can leave the port (hole) open. Other technologies use a retention catheter known as a "plug". It can be a valve system (rubber septum) that assists in pressurized applications with the presence of a self-sealing septum that prevents leakage, maintains the precise pressure in the application process, prevents product loss or waste, minimizes the injection locations, limits the wound, inhibits decomposition and/or infection and, in some cases, accelerates healing. Leaks can occur depending on the application method if a lot of pressure or large application volume is exerted. Therefore, creating tests to determine thresholds for these criteria, such as pressure and strength during the application, depth, and size of the hole on a given crop to prevent leaks, waste, or cracks in the trunk, is essential [15,27,43–45].

(iii)　*Injection methods*: The injection method uses applicators and can be divided into two categories:

(A)　**Pressurized techniques** can reduce treatment time because pressure accelerates the absorption of the applied products. In addition, this technique is more effective for some species of trees that are naturally slower to absorb, especially on cloudy/cold days when translocation/evaporation takes longer. One drawback of using this technology is the possibility of bubble formation (embolism) that can crack the bark and trunk, creating leaks or product rejection when applied under high pressure. Trunk water content and its hydraulic process can be non-invasively monitored through frequency sensors. Studies have shown that the injection of air and dyes can fill many vessels close to the application site with air [46–48]. For this reason, procedures for accurately assessing pressure for a given crop should be implemented to prevent damage related to excess pressure in the trees.

(B)　**Nonpressurized** or depressurized techniques: When the drill cuts the sap-conducting vessels, sap flow as well as water and pressure potential (potential water gradient) exerted in this affected area are stopped; therefore, the absence of an external force impairs the absorption of products in the trunk,

losing translocation efficiency in the leaf evapotranspiration process [15,49]. Depending on the formulation, it may cause the product to precipitate in the ports. According to Kuhns [15], the lack of pressure slows absorption and treatment, requiring a greater volume to be applied, thereby increasing costs. It is important to emphasize that when applications leave an external device, such as a tube or container, the risk of chemical exposure and vandalism may increase as remotion or breakage of the accessory by malicious damage or intentional destruction. For these procedures, when the ports are permanently open, they also become an open wound. Successive treatments using fixed accessories can be associated with necrotic area development. Since the injured tissue is susceptible to the entry of bacteria, it is exposed to the accumulation of water, causing it to rot near this area. Still, over time, in some cases, the trunk/stem can cause severe exudation or damage to the accessory, such as material dryness and breaks inside the trunk. Therefore, leaving a plant with an open wound for endotherapeutic applications over time increases the likelihood of serious problems, such as rotting in the area around the wound and the waste of the applied product [15].

(iv)  *Injection volume:* Classification is related to the volume applied.

**Table 1.** Classifications of endotherapeutic systems.

| Classification Parameters | Description | Commercial Examples |
|---|---|---|
| *Trunk opening* | **Drills:** Most technologies available on the market use perforations with 4 to 10 mm drill bits. This definition is associated with the structure and type of stem of the crop. Drilling above these dimensions is not recommended as it causes major injuries to the trunk. | Arborjet®, Fertinyect®, Arboprof®, Chemjet®, ENDOplant®, ENDOkit Manual® |
| | **Blades:** Opening the trunk without using drills. The technologies that use blades reduce the impact of disruption of vascular tissues, as the sharp spirals of the drill bits do when cutting the tissues during the insertion and removal of the drill. Because it does not form a space for absorbing the applied product, strong pressure is needed to introduce them. This can generate structural damage to the trunk. Depending on the species and climate/season, it may take a long time to introduce products. | Bite®, Arborsytem® |
| *Interface method* | **Plugs:** These represent an important communication between the tree vascular system and product application equipment. When installed, the plugs are stuck/fixed in the bark and/or in very close points and serve as an access point for the application of the product. There are some models of plugs with different diameters on the market. | Biodegradable such as Arborbiokaps® and Medicap, or permanent ENDOterapia Vegetal™ and Arborjet®. |
| *Injection method* | **(A)   Pressurized:** Pressurized technologies exert external pressure on the applicator and force the introduction of the products in the port area inside the trunk. | |
| | (1)   Constant pressure: These technologies propel the products using a determined/continuous pressure that can be from self-dispensing syringes with springs, capsules, or pressurized bags. | Fertinyect®, Mauget®, Chemjet® |
| | (2)   Varied pressure:<br>(a)   **Continuous Flow:** Technologies That also exert a certain pressure during application but which can be adjusted for each plant or culture using cylinders or gas pumps; | Intus®, Arboprof® |

Table 1. *Cont.*

| Classification Parameters | Description | | | Commercial Examples |
|---|---|---|---|---|
| | (b) | **Discontinuous Flow:** | | |
| | | (i) | Technologies that do not show the pressure exerted due to the lack of a measuring system. Depending on the treatment, volume, and type of plant, it is necessary to have more than one application point; the external pressure exerted depends on the type of plant. | Bite®, Arborsystem® |
| | | (ii) | Technologies that allow pressure adjustment through a manual pressure calibration system at the time of application for each plant/species; | Arborjet® |
| | | (iii) | Technologies with an integrated calibration system to determine the pressure and control it during the application, which can be interrupted when the maximum pressure is reached and normalized using an automatic purge system, expelling the air from the system to prevent embolisms. | ENDOterapia Vegetal™ |
| | **(B)** | **Nonpressurized:** The applied product is not introduced into the plant under external pressure. | | |
| | | (a) | **Pipe/catheter:** After port opening, a pipe/catheter is permanently installed inside the tree where the product is applied; | Vita Caule®, SOS Palm® |
| | | (b) | **Gravity:** After the port has been drilled, a system of tubes and accessories is installed from a hanging container above the application point, where the products drain by gravity into the trunk. The only pressure exerted on the product to enter the port is due to gravity. | Medi-ject®, Xyllakill |
| *Injectionvolume* | **Macroinfusion/Macroinjection:** Corresponds to systems where volumes greater than 15 mL are applied to each port. | | | |
| | **Microinfusion/Microinjection:** This is equivalent to applications with a volume of less than 15 mL at each port point | | | |

## 2. Physiology and Plant Morphology

Endotherapeutic applications should be carried out on trunks with good structure and vitality that minimize the impact of performing shallow and small wounds whenever possible. Applying the formulation to the right tissue, considering trunk and species characteristics during active transpiration, and understanding the phenomena and aspects of plant physiology and morphology are fundamental for the success of endotherapy.

The cohesion–tension–adhesion theory initially proposed by Dixon and Joly in 1894 [1] is the most accepted model for explaining the plant's xylemic upward movement of crude sap. Cohesion–tension–adhesion is based on water properties known as capillary action, which explains the movement of the water of the capillary attracted towards regions of lower hydrostatic pressure. Cohesion is the tendency to form an extensive hydrogen bond and mutual attraction between molecules. Adhesion is water attraction to a solid phase, such as the cell wall of plants. Further, cohesion has a high surface tension that causes energy to increase the surface area of a gas–liquid interface. The water is evaporated from the mesophilic cells inside the leaf as surface tension and adhesion on the evaporative surfaces of the leaves pull water through the xylem. The capillarity of the tiny woody vessels formed by water columns with upward movement of the sap is driven by gradients of chemical potential and pressure from the roots to the top of the plants. The functional technique of trees for moving fluid is to drive the sap through negative pressure gradients

generated by water evaporation from the leaves. Pressure from roots alone is insufficient to push water to the top of a tall tree in a continuous flow of soil-plant-atmosphere sap. Thus, the driving force is generated by surface tension on the leaf's transpiration and evaporation surfaces. Photosynthesis (leaves absorb carbon dioxide ($CO_2$) from the stomata to produce sugars) occurs during sap evaporation, converting it into elaborate sap distributed to non-photosynthesized parts. In other words, plants transform $CO_2$ and light into biomass, depending on the efficiency of water use and the rate of transpiration [7,50–55]. Figure 1 illustrates the cohesion–tension–adhesion theory.

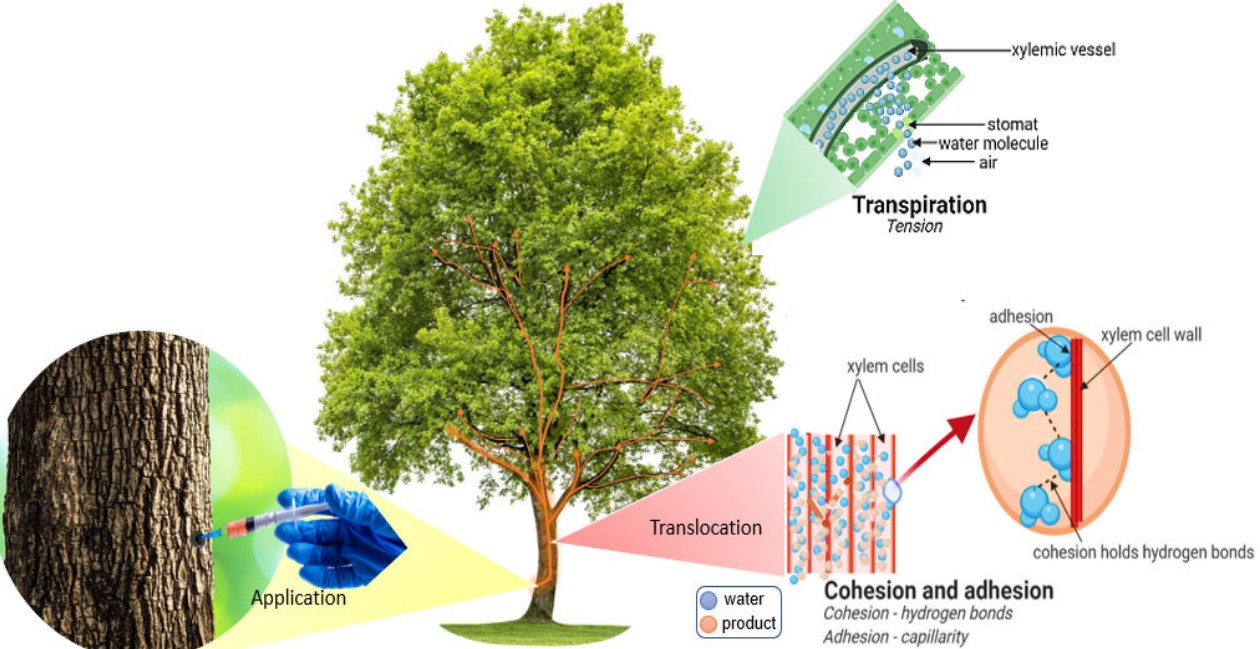

**Figure 1.** Product application by endotherapy followed by translocation from the trunk to the leaves explained by cohesion–adhesion–tension theory.

The sap flow in the trees varies according to the trunk diameter and quantity of water stored in the trunk tissues [56,57]. A thorough understanding of the plant's main morphological, physiological, and anatomical processes allows for improving cultivation and developing GAPs. Likewise, the translocation associated with sap flow may differ between plants of the same variety/cultivar. This is because the translocation–transpiration equilibrium constant in each plant is unique and depends on intrinsic (plant growth and development) and extrinsic factors, which include soil, wind to stimulate perspiration, changes in vapor pressure, temperature, sun exposure, water availability, atmospheric humidity, as well as genetic variability, among other factors [55].

Leaves play a crucial role in the efficacy of endotherapeutic products applied to the trunk by helping in sap translocation. Water is continuously absorbed and lost when carrying out the gas exchanges necessary for photosynthesis, reaching up to 100% of water exchange in a single hour on a hot, dry, sunny day. As translocation is directly linked to transpiration, there will only be translocation of water and mineral salts by xylem and sugars by phloem in the plant with leaf transpiration [51,53,58]. Therefore, the principle of endotherapy considers that products, when applied via trunks, "hitchhike" or are driven by translocation-transpiration balance during sap movement. Applications carried out in warmer seasons result in more efficient treatments because of the direct correlation between higher temperatures and transpiration and, consequently, translocation.

As endotherapy is an endogenous treatment, garnering expertise about trunk types (hardwood or softwood), distribution of sap-conducting vessels, morphological structure of their vascular system (arrangement, size, secondary growth), classification of trees such as

angiosperms (monocots and dicots) or gymnosperms, becomes crucial. Also, depending on the crop, the season should be considered, as there may be senescence and leaf abscission (leaf fall), flowering, and/or fruiting. All of these factors are discussed below based on the papers published in the last 18 years applied to different crops, endotherapeutic equipment, and formulations.

Figure 1 illustrates product application via endotherapy and translocation through the trunk to the leaves based on the cohesion–adhesion–tension theory.

## 3. Endotherapy in the Last 18 Years

A bibliographic search was carried out using the terms *trunk injection tree*, *infusion*, *or pressurization in plants*, using the ISIS Web of Knowledge database from January 2005 to December 2022. The publications were compiled and evaluated according to the year, culture, trunk opening, injection method, interface method, target compound, product class, brand, statistical tool, analysis, and journal. Additionally, these research papers can be consulted/filtered through an Interactive Table (Dashboard) in Microsoft Office Excel for Windows® using data segmentation attached as Supplementary Material according to the References as shown in Figure 2.

Figure 3 evaluates the progress of endotherapeutic techniques in the last fifteen years. The crops were grouped into four categories: (1) Deciduous trees; (2) Conifers; (3) Fruits trees, and (4) Ornamental trees. Figure 3A relates the number of articles published in the same time span, and Figure 3B represents the percentage of articles published grouped into the categories mentioned above. Crops included in endotherapy studies, with emphasis on high-value fruit plants, are detailed in the Interactive Table. The plants were classified into categories to help the discussion of this review.

Endotherapy studies were on five tree groups: deciduous, coniferous trees, fruit trees, and ornamental trees, with 11%, 24%, 11%, and 54% of studies in each of the respective categories.

### 3.1. Crops

Citrus trees were the fruit plants having the most significant number of publications involving different species, varieties, and/or cultivars. These varied among orange, lemon, lime, pomelo, and grapefruit. Table 2 contains information about all fruit species, cultivars, and references studied. After citriculture, the fruit tree with the most publications was the apple tree, with different varieties/cultivars, followed by nut species and other fruit species, with the lowest number of publications.

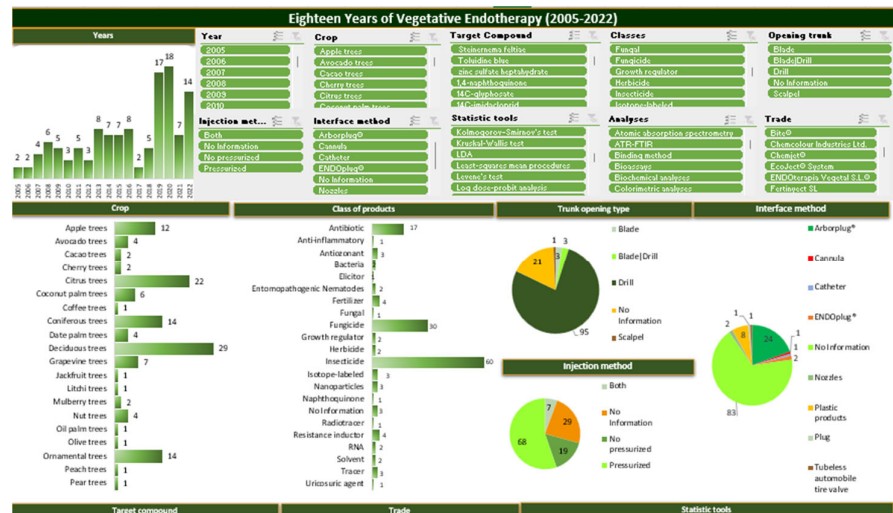

**Figure 2.** Template for the Interactive Table to be consulted and available for download in the Supplementary Material.

**Table 2.** Results of all fruit species, cultivars, and references studied in endotherapy were found using the ISIS Web of Knowledge database from January 2005 to December 2022.

| Classifications | Species | Cultivars/Scientific name | References |
|---|---|---|---|
| Citrus trees | Orange trees | *Citrus sinensis* | [59–61] |
| | | *Citrus sinensis* (L.) Osbeck on Swingle citrumelo (Citrus paradisi) Macf Duncan grapefruit x *Poncirus trifoliata* (L.) Raf | [62–65] |
| | | *Citrus sinensis* L. Osbeck | |
| | | Hamlin sweet orange on Swingle citrumelo rootstock | [60–68] |
| | | Valencia orange trees | [69] |
| | | *Citrus sinensis* L. cv. Valencia | [70] |
| | | Hamlin sweet orange (*C.* x *sinensis*) | [71] |
| | Grapefruit trees | *Citrus paradisi* Macfad | [59,61,72] |
| | Lime trees | *Citrus aurantifolia* Swingle | [49,73,74] |
| | | Mexican lime and clementine mandarin cv. Nour grafted to Carrizo rootstock | [22,75] |
| | Pomelo trees | *Citrus maxima* | [16] |
| | Mandarin trees | *Citrus nobilis* Loureiro onto volkameriana rootstock (*C. volkameriana Pasquale*) | [76] |
| Apple trees | | *Malus domestica* Borkhausen | [12,32,77,78] |
| | | *Malus pumila* Miller | [10,79] |
| | | *Malus domestica* Borkhausen cv. Red Delicious | [80,81] |
| | | *Malus domestica* Borkh. cv. Gala | [30] |
| | | *Malus domestica* Borkh. cv. Mac Spur | [12,81] |
| | | No specific cultivar | [82,83] |
| Nut trees | Almond trees | *Prunus dulcis* | [33] |
| | Macadamia trees | | [84] |
| | Chestnut and walnut trees | | [85] |
| | | Chestnut variety Marrone di Chiusa Pesio | [86] |
| | Persian walnut trees | *Juglans regia* L. | [87] |
| Grapevine trees | | | [19,69,78,88–91] |
| Cherry trees | | | [92] |
| | *Prunus cerasifera* and *Prunus* spp. | | [13] |
| Avocado trees | | | [9,93–95] |
| Cacao trees | | | [96,97] |
| Coffee trees | | | [98] |
| Coconut palm trees | | PB 121 coconut hybrid | [58,99–102] |
| | | | [5,103] |
| Litchi trees | | | [104] |
| Olive trees | | | [105] |
| Peach trees | | | [106] |
| Date palm trees | | | [107–110] |
| Pear trees | *Pyrus communis* L., var. Bartlett | | [33] |
| Mulberry trees | | | [111,112] |

Trees play a crucial role in urban centers for their role in climate stability by increasing atmospheric humidity when carrying out gas exchanges. Many of these trees are grown in unfavorable environmental conditions, which can result in a high rate of disease and pest attacks. Endotherapy is preferable to pesticide spraying as it reduces the risk to the applicator, the human population, and the environment [113,114]. Tree health in urban settings and forests has been a concern of environmental managers and an object of different studies regarding the diagnosis, protection, and remediation strategies. All deciduous tree species/cultivars studied are listed in Table 3.

**Table 3.** Results of all deciduous tree species, cultivars, and references studied in endotherapy found using the ISIS Web of Knowledge database from January 2005 to December 2022.

| Classifications | Species | Cultivars/Scientific Name | References |
| --- | --- | --- | --- |
| **Deciduous trees** | Ash | *Fraxinus* spp. | [29,115–117] |
| | White ash | *Fraxinus americana* | [11,27] |
| | Common ash | *Fraxinus excelsior* | [21–23,118] |
| | Green ash | *Fraxinus pennsylvanica* Marsh. | [11,27,44,119] |
| | Poplar and Ficus | *Populus, Ficus* | [115] |
| | Coast live oak | *Quercus agrifolia* | [113] |
| | Cork oak | *Quercus suber* L. | [42] |
| | OakEnglish oakPlane treesLondon plane | *Quercus* spp.; *Quercus robur*; *Platanus* spp.; *Platanus acerifolia* | [13] |
| | Holm oak | *Quercus ilex* | [3,120] |
| | Pedunculate oak | *Quercus robur* L. | [82] |
| | Horse chestnut | *Aesculus hippocastanum* L. | [13,121–124] |
| | Black olive | *Bucida buceras* L. | [125,126] |
| | Paper birch | *Betula papyrifera* Marsh. | [46] |
| | Norway maple | *Acer platanoides* L. | [127] |
| | Queensland Brush Box. | *Lophostemon confertus* | [128] |
| | Elm | *Ulmus americana* | [129] |
| | Wiliwili | *Erythrina* spp. | [130] |
| | Lead | *Leucaena leucocephala* | [131] |
| | Black walnut | *Juglans nigra* | [132] |
| | Black cherry | *Prunus serotina* Ehrarth | [17] |
| | | *Archontophoenix cunninghamiana* (H.Wendl.) H.Wendl. & Drude; *Bauhinia picta* (Kunth) DC.; *Caesalpinia pluviosa* DC.; *Eriobotrya japonica* (Thunb.) Lindl.; *Ficus benjamina* L; *Fraxinus chinensis* Roxb.; *Handroanthus chrysanthus* (Jacq.) S. O. Grose; *Jacaranda mimosifolia* D. Don; *Lafoensia punicifolia* DC.; *Lagerstroemia speciosa* (L.) Pers.; *Pithecellobium dulce* (Roxb.) Benth.; *Roystonea regia* (Kunth) O. F. Cook; *Spathodea campanulata* P. Beauv.; *Terminalia catappa* L.; *Syzygium malaccense* (L.) Merr. & L. M. Perry. | [114] |

Studies have shown that conifers are susceptible to various pest pressures [2,41,133,134] and nematodes [135–137]. Furthermore, as tree height can hinder the use of foliar product applications, endotherapy is a viable alternative. The evaluated conifer species/cultivars are listed in Table 4.

**Table 4.** Results of all conifer species, cultivars and references studied in endotherapy found using ISIS Web of Knowledge database from January 2005 to December 2022.

| Classifications | Species | Cultivars/Scientific Name | References |
|---|---|---|---|
| **Coniferous trees** | Pine | *Pinus massoniana* | [136] |
| | | *Pinus thunbergii* | [135,138] |
| | | *Pinus pinaster* Aiton | [137] |
| | | *Pinus ponderosa, Pinus contorta, Picea engelmannii* | [133] |
| | | *Pinus densiflora* | [139] |
| | | *Pinus pinea* L. | [140] |
| | Japanese Cedar | *Cryptomeria japonica* | [141,142] |
| | Eastern Hemlock | *Tsuga canadensis* Carrière | [143] |
| | Hemlock | *Tsuga* spp. | [41] |
| | Grand fir, Douglas-fir, alpine fir | | [2] |
| | Cedar of Lebanon | *Cedrus libani* | [13] |
| | Norway spruce | *Picea abies* (L.) Karst. | [134] |

Since ornamental trees are typically planted in urban areas, endotherapy has been considered an important technique to prevent environmental contamination [81]. This work describes the species considered ornamental trees in Table 5.

**Table 5.** Results of all ornamental tree species, cultivars, and references studied in endotherapy found using the ISIS Web of Knowledge database from January 2005 to December 2022.

| Classifications | Species | Cultivars/Scientific Name | References |
|---|---|---|---|
| **Ornamental trees** | Canary Island date palm | *Phoenix canariensis* | [43,144] |
| | Sweet olive | *Osmanthus fragrans* | [145] |
| | Chinese banyan | *Ficus microcarpa* L. | [146] |
| | Plane | *Platanus* × *acerifolia* (Aiton) Willd | [147] |
| | *Magnolia virginiana* L. | | [148] |
| | Willow | *Salix matsudana* cv. 'Pendula' | [149] |
| | Palm tree | | [13] |
| | Tobacco *Nicotiana benthamiana* | | [150] |
| | Japanese cherry trees | | [151] |

*3.2. Products Used in Endotherapeutic Treatment*

Agricultural products include (i) active ingredient, a substance that has an action/effect on the target; (ii) inert products, non-reactive substances in the mixture; and/or (iii) adjuvants, substances that can improve formulated performance. The adjuvants in these formulations aim to increase the solubility of the active ingredient in the sap and enhance translocation. Some studies have added adjuvants such as 2-(2-ethoxyethoxy)ethanol and acetic acid [132]; acetic acid, acetone, ammonium nitrate, hydrochloric acid, nitric acid, potassium hydroxide [118], and salts (urea, potassium chloride, and sodium chloride), citric acid, and organo-silicones (Break-thru® and Silwet-L77®) [99].

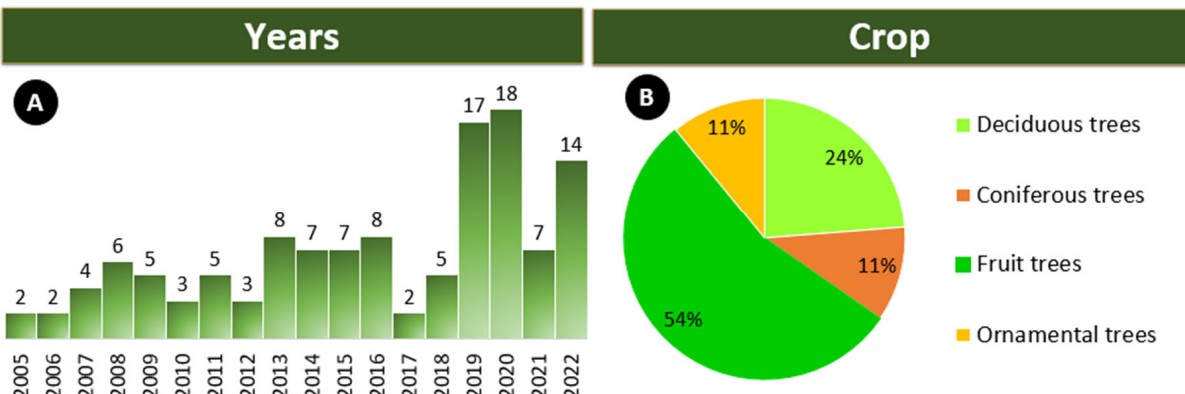

**Figure 3.** (**A**) The number of publications per year focusing on endotherapy (trunk injection tree, infusion, or pressurized) in plants. (**B**) The percentage of articles published is grouped into four main categories (ISIS Web of Knowledge, January 2005 to December 2022).

When using a formulation in endotherapy, it is essential to certify how the product will be used, the concentration of active ingredients, solution viscosity, potential for systemic translocation, and active ingredient stability, among many other characteristics. The systemic behavior of a product is related to the ability to translocate within the plant and the target location. Therefore, the systemic potential is a determining factor for endotherapeutic treatment [1,6,11]. Active ingredient translocation efficiency within the plant's vascular system depends on its physicochemical properties such as molecular mass, water solubility, lipophilicity (Kow), polarity (pKa), and pH of the solution, in addition to the presence of other ingredients that compose the formulation. To Berger and Laurent [1], commercial formulations designed for spraying are not necessarily compatible with vascular transfer. Therefore, the formulations must modify the physicochemical properties of the active substances to improve their distribution within the tree, such as water solubility and low Kow.

The formulation is one of the biggest challenges in studies involving endotherapy. Specific products for endotherapy are limited, be it in the variety of active ingredients or formulated products. The companies themselves that develop products for this purpose are also few. Although commercial products have the same main active ingredient, formulations vary between manufacturers, which may influence the products' applied translocation efficiency. Therefore, dosage specifications considering the age of the species/cultivar, trunk diameter, plant height, and application interval; studies of maximum residue limits for fruit plants are still in progress [48,58]. Some commercial products with formulations for endotherapeutic applications include J.J. Mauget Co., Arborjet®, and ArborSystem® [10,94,116,130].

An estimation of dosage (injection rate) and the number of ports based on the diameter at breast height (DBH) [113,134], as well as the tree's unique trunk diameter at one-foot height (DFH) [30], has been reported. For example, an avocado orchard had a DBH of approximately 50 cm, and the authors calculated, from the active ingredient of fungicide, 1.1 mL cm$^{-1}$ [18]. Acephate was injected with concentrations at 0.0 for the control sample, 0.25, 0.50, and 1.00 g cm$^{-1}$ tree diameter measured at DBH in *Lophostemon confertus*, a commonly planted street tree in Australia [128], and azadirachtin-treated ash trees at a rate of 0.2 g azadirachtin cm$^{-1}$ at DBH [117]. However, no recommendations cover dosage for many other active ingredients and cultures.

Many studies have used active ingredients pre-formulated for research as they are more accessible and economical than buying technical quality products from suppliers. These studies have also reported that highly diluted products performed the best in terms of faster uptake and establishment of effective concentrations [93]. Product formulation and concentration raise concerns about xylemic translocation functionality and tissue function impairment after several years of using successive endotherapeutic applications [12].

In some cases, high concentrations may have a phytotoxic effect, causing necrosis and discoloration of vascular tissues ([19,21,86]. However, other studies report that low concentrations are ineffective in pest control. Some studies have overcome this problem by increasing product concentration in the formulation [94]. In cases of disease and/or pest control and management, choosing the most efficient product to be applied requires evaluating the infestation and affected tissue (parenchyma, xylem, or phloem), classifying the insects (sucking, chewing, or boring) and the diseases caused by viruses, fungi or bacteria (ecto- or endoparasites) [1].

Several product classes have been applied by endotherapy. Twenty classes were identified containing products used exclusively for endotherapy, and the insecticides, fungicides, and antibiotics most used were: imidacloprid, emamectin benzoate, abamectin, oxytetracycline, azadirachtin, dinofeturan, penicillin G, phosphorus acid, and thiamethoxam, as shown in Figure 4.

Although endotherapy has been mainly used to apply insecticides, fungicides, and antibiotics, several studies have shown the applicability of other product classes. This information can be consulted using the Excel interactive dashboard (Supplementary Material). Experiments were carried out with the objectives of preventing and controlling pest outbreaks and using markers to visualize translocation behavior as tracers, including isotope-labeled and radiotracers. Some new product strategies are being developed to replace antibiotics and pesticides. Extensive application of pesticides and antibiotics without criteria can pose a major problem in creating resistance to pathogens [72]. Currently, with the number of new active ingredients available, bioformulations are created to control the pathosystemic problem with endotherapeutic applications. Vegetable hormones such as jasmonic acid, ethylene, salicylic acid, and gibberellin have been used in agriculture to regulate plant growth, abiotic stress tolerance and to induce defense response resistance against pathogenic infections [83,152]. These have been applied by endotherapy.

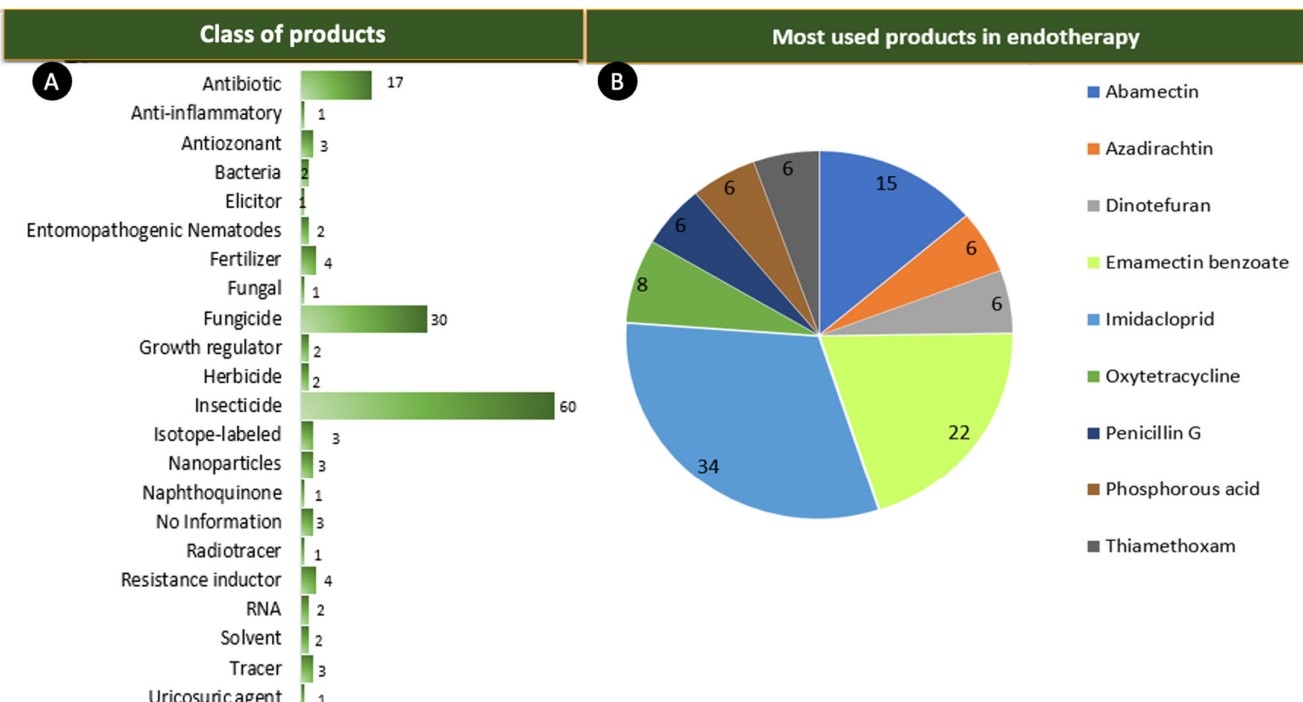

**Figure 4.** The number of publications emphasizing the classes of products applied using endotherapy. (**A**) Classes of products used; (**B**) most used products. (ISIS Web of Knowledge, January 2005 to December 2022).

Recent studies in nanotechnology present more possibilities for endotherapy as products based on nanoparticles are strong candidates to be more potent agents than active

ingredients currently used as antibiotics to decrease bacteria in plants [49,75]. Other types of bioinsecticides and bactericides for applications in sustainable agriculture from an eco-toxicological perspective were used, such as neem oil (azadirachtin) [102,117] and garlic oil (allicin) [118]. A study was conducted using aquatic and terrestrial microcosms and proved that using azadirachtin for endotherapy probably does not pose a significant health risk to aquatic or terrestrial invertebrates when leaves treated with insecticide fall from trees. Still, the authors suggest that azadirachtin may be an option for a systemic insecticide, especially in ecologically sensitive environments, such as conservation areas, riparian forests, water springs, and wooded wetlands [117]. Research has proven that the systemic translocation of cinnamon and spearmint essential oils as biopesticides applied by endotherapy to apple trees resulted in the finding of targeted and untargeted volatile organic compounds in leaves. This research highlights the potential induction of systemic acquired resistance (SAR) using products that were not phytotoxic [153]. The promising area uses new phyto-chemicals with the discovery of nematicidal activity as biopesticides from different essential oils and their volatiles [154]. Also, a new class of stable antimicrobial peptides could inhibit *Candidatus Liberibacter* asiaticus (CLas) infections that cause one of the most devastating citrus diseases in the world [155].

## 4. Evaluation of Different Endotherapeutic Treatments

To select endotherapy as the application method, it is essential to identify which technique is most suitable for each culture. For example, equipment used for apple or avocado trees may have to be adapted for an endotherapeutic application in palm trees. Since some systems may not meet the requirements and conditions of a given culture, hiring well-trained professionals to safely and efficiently perform applications is crucial.

After the crop, agricultural products, and seasonality have been evaluated and defined; the following steps are trunk opening to use the interface method that indicates the injection method (delivery pressure) for the application of endotherapeutic equipment. Treatment effectiveness depends on many factors, including the anatomy and physiology of the trees, trunk health condition, environmental conditions, the product formulation applied, and the type of endotherapeutic technique used [4,78,118]. The graphic data related to the type of trunk opening, interface method, and injection are represented in this text as a percentage to facilitate the discussion. The interactive Table in Supplementary Material presents the number of publications to assist the interactivity and interpretation of the results.

The trunk opening is one of the biggest challenges facing endotherapy due to the damage caused, which leads to controversy when the technique is applied. However, these injuries were due to early mistakes, which, in many cases, have been rectified, and now trees can "heal" these injuries. It is important to note that in trees, the injured/broken vegetal tissues do not regenerate and are not replaced. The peripheral cells and damaged area create barriers, protecting the healthy area, and this process is called compartmentalization or sealing. Yet, this is not the case in palms, as instead of containing layers of tissues as in other trees, palms are composed of countless vascular bundles distributed throughout their length. Thus, trees require more targeted techniques than palms due to vascular cambium and sapwood. The center of the stem shelters the "heart" of the trees, known as the heartwood, which consists of a dead zone and, when injured, is subject to invasion of microorganisms. Palm trees, on the other hand, are affected when the apical meristem, their only growth point, is reached. This justifies the difference in depth as there are shallower holes in trees than in palms [8].

Different forms of trunk openings, interface methods, and injection methods have been used, and Figure 5 shows the percentage of all resources approached by endotherapeutic systems. Few trunk opening type options were used, owing to a lack of alternatives and limited use of the perforations with the aid of a drill; the majority corresponded to 77% of the papers found, followed by 21% that needed to be more informative. It was noted that many studies did not specify the dimensions of the drill bits. This information could be relevant because it can impact the diameter and depth of the hole (port). Some works

considered the application site as a port, which can specify the number of ports per tree, and the influence of the direction, as cardinal direction (N, S, E, W) orientations [4,12,132]. Few articles used an alternative strategy, such as blades [13,77,118,132,134], as shown in Figure 5A. Two companies offer blades, Wedgle® Direct-Inject tree injection™ from ArborSystems® and Bite® from De Rebus Plantarum. Al-Rimawi and collaborators used a scalpel to open the trunk since their experiments used only 3-month-old citrus seedlings, so another opening instrument would not be viable [73].

The number of ports per tree generates controversial points of view due to the number of wounds sustained on the trunk. In some trees, the ports can cause mucilaginous exudates or cracks on the trunk, as well as callus formation and healing for 6/7 months in the bark after treatment were the physiological responses of trees [121]. Assessments suggest that four application ports allowed a more uniform spatial distribution of imidacloprid than using only one or two ports. However, no further advantage was observed with eight ports on apple trees. Furthermore, the temporal distribution of imidacloprid was significantly non-uniform, but regardless of this variable, observing the product in the canopy showed it was long-lasting [12]. In addition, Hu and Wang [67] evaluated two ports to achieve optimal canopy distribution. Other studies have reported that a single application port showed the best results in minimizing damage by applying two different products in the same season when controlling insects and diseases in apple trees [80]. Additionally, limiting the number of ports decreases physical damage to trees [149]. A few studies have reported that both the application systems and the drill bits, tools, and needles should be sanitized between each injection to avoid infection or microbial contamination, in addition to disinfecting the ports with a suitable fungicide [32,121]. Some articles reported closing the ports using wood, stoppers [123], and even an alternative using grafting wax not to expose the open wound [147]. The ports that used these materials only to close the hole were not counted as an interface method.

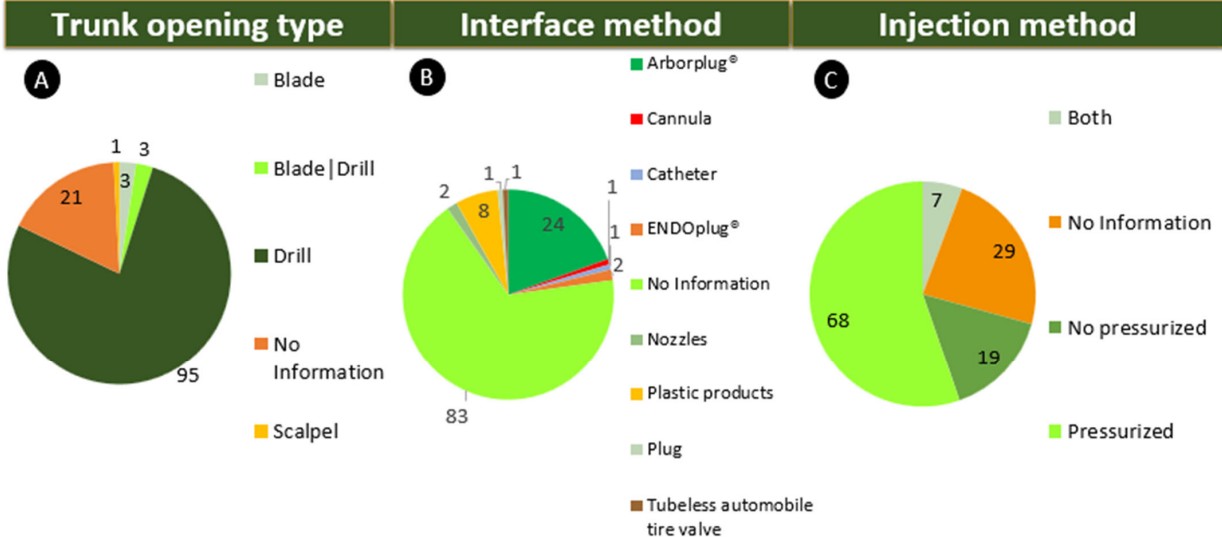

**Figure 5.** The number of publications showing: (**A**) trunk opening type, (**B**) interface method, and (**C**) injection method using endotherapy. (ISIS Web of Knowledge, January 2005 to December 2022).

Figure 5B presents the results of interface methods used on the trunk for product delivery systems applied as accessories to aid in pressurizing the system, such as the plugs from Arborjet® found in 20% of the papers, followed by ENDOterapia Vegetal® [90,111], nozzles [84,122], tubeless automobile tire valves [147] and among others. Plastic products were also used as an interface method (7% of the papers) and provided pressurized systems. However, a significant number of papers (67%) were not informative in this respect and failed to specify which interface method was used. Interface methods can be easily consulted in the interactive Excel dashboard.

Aćimović and collaborators carried out a study that evaluated port injuries generated from endotherapeutic treatments using drills or blades, which included the following metrics: healing rate, port diameter, bark crack size, and port depth from the use of commercial plugs, blades or no plugs (unsealed drill port), and monitoring wound behavior over time. The results showed that plugs delayed healing with calluses and increased depth over time and that the results were significantly faster when using a blade, followed by results using unsealed drill ports [77].

Figure 5C shows which injection methods were used: 55% of the treatments were pressurized techniques, 15% were non-pressurized techniques, 24% did not provide information concerning the method used, and only 6% tested both (pressurized and non-pressurized) in the same work. Some examples of pressurized systems used were spring-loaded syringes, pressure applicators, air cylinders/tanks, and capsules. In the last 18 years, the pressurized endotherapeutic technique mentioned in most publications, corresponding to almost half of the articles, was ArborJet®, using different innovative technological systems. Twelve other companies with different technologies had their products described in papers and were included in the Excel interactive dashboard. In other works, 15% used simpler non-pressurized systems, highlighted by a Xyllakill container proposed by Stephano-Hornedo and collaborators [49]. In some cases, homemade manufacturing was adopted because it was a cheaper and easier method of operation without the need for specialized equipment [128].

In papers that evaluated both injection methods, findings showed that the pressurized technique required less time for absorption/translocation of the product in horse chestnuts, 10 min for healthy plants and 5 h for sick ones. For the non-pressurized method, the natural absorption was 60–70 min for healthy plants and 15 h for sick ones. This study was only based on evaluations in field trials, and there was no significant difference between the two methods and did not include a specific analysis of plant tissues that evaluated concentration or the persistence of the product inside the plant. In addition, the endotherapy treatment showed insect control for over two years, with considerable cost savings [121]. The study that evaluated silver nanoparticles' performance in citrus tree diseases used two injection methods at the same concentration, non-pressurized, Xyllakill, and pressurized, Arborjet®. The high concentration associated with high pressure suggested an agglomeration of nanoparticles in the vascular tissues, which hindered the effectiveness of bacterial elimination. However, these results were significantly better than those using spraying. In later studies, the authors applied diluted solutions using pressurized techniques [49]. Other inconclusive studies did not discuss the difference between pressurized and non-pressurized treatments [132,134,144,149].

Some studies compared different commercial technologies to evaluate treatment efficiency. In the first experiment, Xu and collaborators evaluated three systemic insecticides and imidacloprid-controlled gall wasps in Wiliwili trees the best. In the second experiment, Sidewinder precision tree injector, Mauget® 3-mL capsules, ArborSystems Wedgle Direct Inject, and an Arbor-Jet Tree I.V. were used for endotherapeutic applications, as well as a test using root zone drench. The results showed superior efficiency in the treatments using the two products and applying the technology via Arborjet®. Endotherapy results were detectable one year after applications, varying according to the technique and formulation used. Applications via soil drenching did not present satisfactory results [130]. Cowles and colleagues also compared treatments using imidacloprid formulations with applications according to each manufacturer: Arborjet® VIPER system, Mauget System, and Wedgle Direct-Inject Tree Treatment System compared some soil application modalities to control the woolly adelgid hemlock in forests. The results found that applications via soil were more satisfactory than endotherapy, contrasted with those by Xu and collaborators [41]. This discrepancy has led to some hypotheses: (a) crop type and pest breed that were treated; (b) how the applications were conducted. More recent studies have shown superior results and economic and environmental advantages over direct soil treatments since endotherapy applications tend to be longer-lasting. Also, in these evaluations, soil type and moisture,

tree location, plant tissue type, and target pest feeding must be considered [12,132,144]. A citrus tree study using nanoparticles in endotherapeutic applications showed more satisfactory results than application on the petiole, root, branches, and leaves [75].

Studies have shown some relevant approaches to managing product resistance using endotherapy, mainly for pesticides, such as rotating different active chemicals in rows of trees or individual trees subject to selective pressure on an insect population to prevent resistance to applied products from developing [12,28,144]. Another study showed a reduction and control of nematode infestation in trees that were not treated after the application of insecticide in a limited forest area [149]. This study suggests the possibility of further testing with the intercalation of trees for randomized application of products, reducing the costs and time of operation in the field.

Most papers show that the experiments used water or distilled water as a control method. This procedure is essential and must be integrated into the protocols to evaluate whether the plants in a given area are cross-contaminated [118]. Application time for endotherapy has better translocation results when applied in summer when transpiration rates are the highest and substances rarely move down the trunk [80,106,116]. However, the application will depend on the treatment type to define the applied product as the dormancy period before sap flow [19] and the senescence period [117]. For pressurized methods, the best applications were obtained in the early morning (8:00–11:00) and then in the late afternoon (18:00–20:00) [118].

## 5. Analysis after Endotherapeutic Applications

After the endotherapeutic applications were submitted to established and standardized conditions, the plants were subjected to analysis based on observations, and monitoring in field trials, green/glasshouses, and/or using analytical equipment aided by statistical tools. Many authors performed more than one type of analysis, which, in most cases, corroborated by using different statistical tools to analyze results. In the last 18 years, studies using field/semi-field trials accounted for 32% of the analyses performed, followed by 20% of residue analyses, 13% of bioassays, 9% of PCR, and 6% with green/glasshouse trials. The sum of all other types of analysis is 20%, as shown in Figure 6.

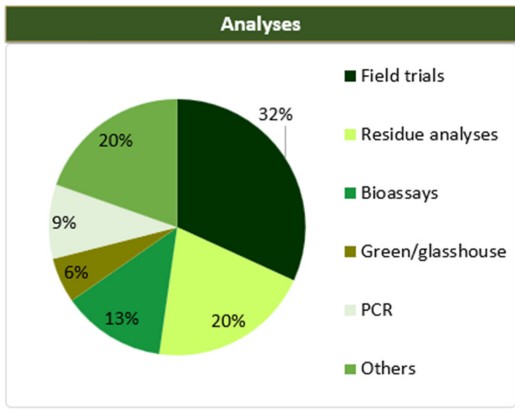

**Figure 6.** Percentages of publications emphasizing the analysis types after endotherapeutic treatments in the last 18 years. (ISIS Web of Knowledge, January 2005 to December 2022).

Examples of evaluation methodologies and criteria considered in field trials were based on defoliation levels [120,145], levels of severity of progressive deterioration of the affected crown [41,114]; insect population mortality data [41]; and visual assessment of disease severity [82]. The number of stem galls, stem and leaf insect infestation levels, visual ratings on new shoot emergence, overall tree health [146]; evaluation of foliage density of the mistletoe [115]; pest attacks [126]; gall density and staining [125]; and larval mortality [145]; larval penetrations and efficiency of the treatments [87]; reduction in exit holes of *Xylotrechus chinensis* [112] were evaluated. In some studies, the experiments

were conducted in a glass/greenhouse to obtain better investigation control, not exposing products to the environment or putting pollinators at risk. In other cases, a complement to the field trial was used [16,40,59,62,72,73,78,85,105,129,130]. Other authors point out that applying a protocol or methodology in a glass/greenhouse uses unrealistic conditions, thus making it difficult to compare to conclusive results in field conditions [21–23].

Qualitative or quantitative residue analyzes used vegetative tissue from trees such as bark, roots, leaves, flowers, and fruits to determine the presence/absence of the compound. Most studies used concentration. Quantitative analyses consider the concentration of residues in parts of the plant, allowing measurement of how much of the analyte was translocated after endotherapeutic application, different from the qualitative analysis that did not determine concentration values. One of the determining factors of the analysis type is the availability of the appropriate analytical instrument in the laboratory. Chromatographic analyses use the physical–chemical characteristics of the molecules, such as polarity and volatility, among others, usually after the extraction of the analytes. Fungicides on leaves and apple fruit used gas chromatography coupled to mass spectrometry (GC-MS) and liquid chromatography coupled to mass spectrometry (LC-MS) [81]; fungicides and insecticides in the coconut stem used liquid chromatography coupled to tandem mass spectrometry (LC-MS/MS) [58,99–103,156]; fungicide in roots and trunk was evaluated by GC-MS [106]; uricosuric agent and anti-inflammatory compounds in leaves, roots, fruit peel, fruit juice samples used LC-MS/MS [59]; bioactive constituents present in plants using GC-MS and HPLC [90]; insecticide in avocados was analyzed by GC and LC, although the authors did not mention the type of detector used [94]. Enzyme-linked immunosorbent assay (ELISA) was used for quantitative and qualitative analysis: insecticide in avocado leaf tissue [9,94]; insecticide in Wiliwili tree leaves [130]; insecticides in avocado leaf and fruit [93]; antibiotics in petiole, midrib, bud or root tissue all used double antibody sandwich-enzyme linked immunosorbent assays (DAS-ELISA) [84]. Inductively coupled plasma mass spectrometry (ICP-MS) was used in quantitative and qualitative analysis of inorganic compounds: silver (Ag) concentration in citrus trees [74]; cesium (Cs) concentration in Japanese Cedar [141].

Some studies carried out bioassays that included the inoculation of flowers and subsequent disease evaluation, shoot blight severity control [30]; test for nematicidal activity [135]; insect performance [123]; in vitro tests using agricultural products [118]; count of individuals (adults and nymphs) [147]; in vitro activity [106]; stock colonies and experimental insects [43]; and diet bioassays [93]. Quantitative real-time reverse transcription-polymerase chain reaction (qRT-PCR) has been used as an indirect determination and for bacteria quantification 'Ca. L. asiaticus' in citrus trees [49,59,66], as well as for quantitative-PCR (qPCR) [69]. This was also used as analysis in the same bacteria and crop [16,20,40,61,62,67,72]. qRT-PCR was used in analyses of the bacteria *Erwinia amylovora* in apple trees [30]. Real-time quantitative-PCR evaluated transcriptional changes and downregulated transcript levels of some genes in pine trees [136] and PCR to analyze *Xylella fastidiosa* in almond trees [84].

Other equipment and different types of techniques used for the evaluation of endotherapeutic applications can be consulted in the Excel interactive dashboard.

## 6. Challenges and Advances

Endotherapy has demonstrated its utility in different cultures, using several classes of products with satisfactory results and numerous environmental advantages, such as protecting water sources, workers, and the communities close to these areas. Furthermore, it was shown that endotherapy went beyond treatments for diseases and pests and could be used as nutritional deficiencies, growth regulators, and resistance inducers. Endotherapy involves complex factors that have not yet been fully elucidated, which hinders effective practice in all circumstances, showing that this task requires more investment and could take years for conclusions and possible solutions. Thus, further studies should be carried out.

Many advances have been made so far, and the greatest expectation is that endotherapy will surpass itself in the coming years as new analysis technologies can certify the interpretation of results obtained. Compared with conventional application techniques, the endotherapy is still small, but the sector has been expanding with a high potential for results. A growing number of publications has shown interest and progress in the following aspects: (a) the progressive application of pressurized techniques; (b) the use of different classes of products for different contexts (diseases, pests, deficiencies, resistance); (c) rotation strategies in the application of different active ingredients and modes of action in an area; (d) use of analytic technologies that monitor and evaluate the translocation efficiency aided by statistical tools.

For the coming years, the challenges for vegetative endotherapy are that the technique will be recognized by government agencies with public laws and policies that establish safe dosage levels using less harmful and more appropriate products through the use of new formulations and bioformulations, as well as the development of new molecules using nanotechnologies, to ensure food security. In addition, producers find any solution that is robust, simple, and easy to use in the field at an affordable cost attractive. Criteria developed and used in conventional applications as spraying and application via soil, should not be applied or recommended for endotherapy, such as the interval between applications, harvest, post-harvest handling, dosages, grace period, safety interval, climatic conditions, and solution dilution.

After assessing the advances in endotherapeutic systems and designing challenges that can improve existing techniques, we have pointed out some areas of improvement for the viability and expansion of endotherapy, such as:

(1)  Development of new technologies and tools to open ports that are less invasive in the tree trunks, especially for palm tree stem (Figures 7 and 8). Some technologies, such as the blades mentioned in this review, are manual and are not practical for hardwood and may have difficulties introducing products such as Bite Infusion® for coconut palm trees.

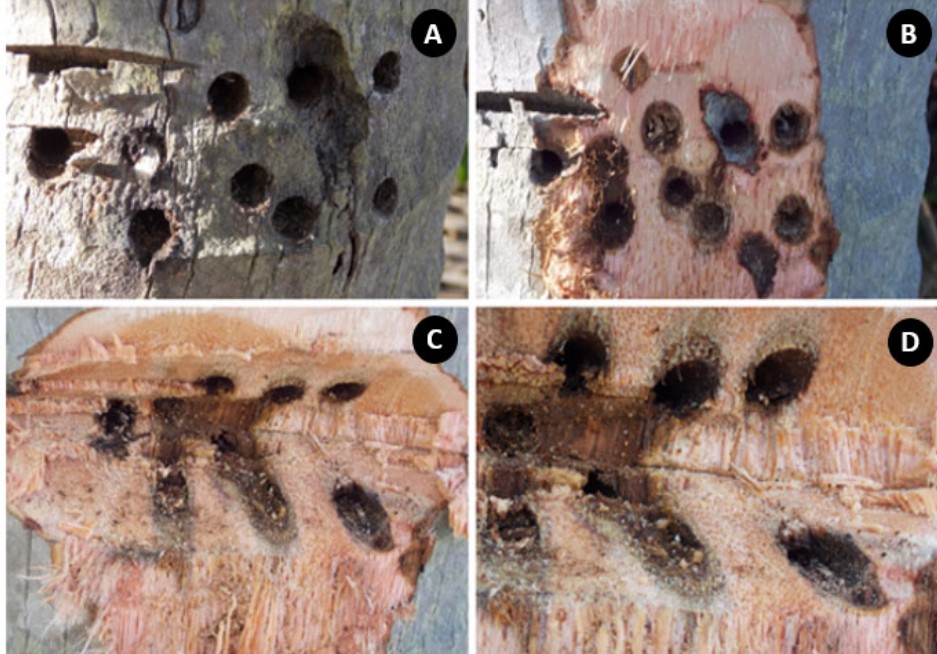

**Figure 7.** Result of some holes made with the aid of a hand drill without applying pesticides. (**A**–**D**) Presents a sequence of pictures with the deepening of the cut to visualize the damage to the trunk in the coconut palm. Source: Reprinted/adapted with permission from Ref. [35]. 2016, Jordana Alves Ferreira.

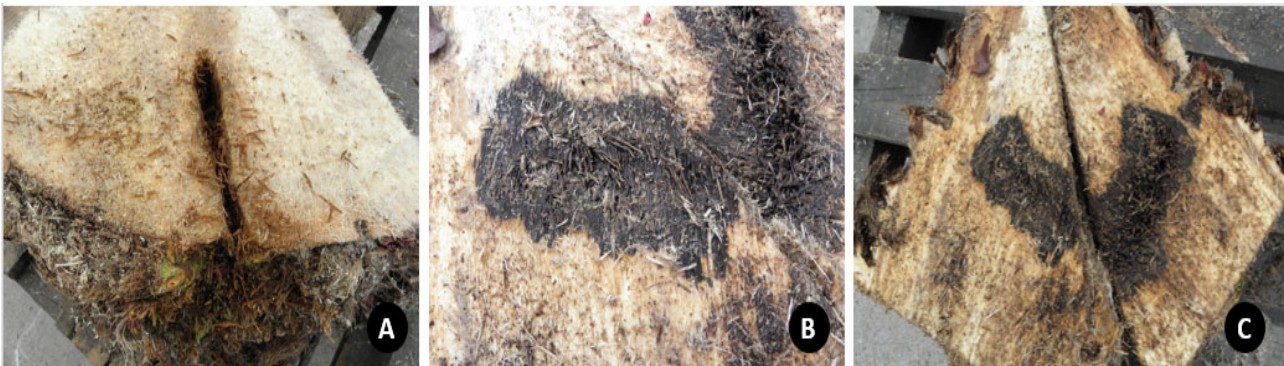

**Figure 8.** Result of the trunk after using the drill in a palm tree (*Phoenix canariensis*). (**A**) Before opening the trunk; (**B,C**) Assessment of trunk necrosis at the point where the drill was used. Source: Collection of the authors' pictures.

(2) Use technologies that avoid excess pressure that can cause the bark to crack (Figures 9 and 10).

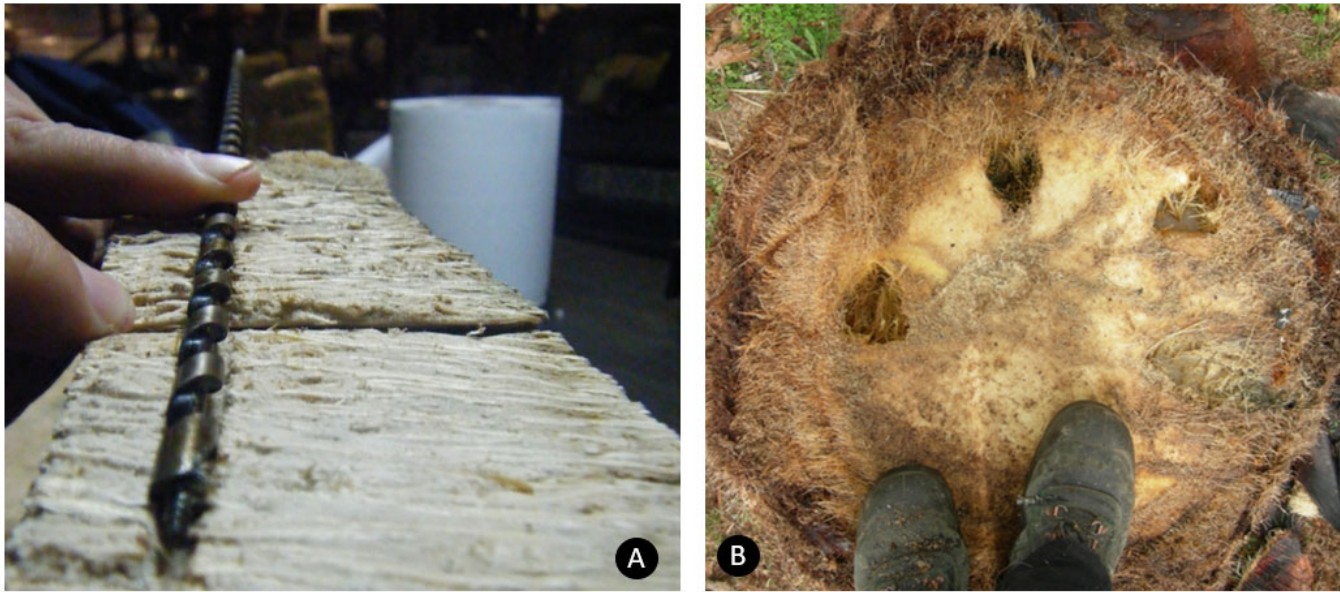

**Figure 9.** Results of the excess pressure at the injection point during the application of the products in *Phoenix canariensis*. (**A**) Using a 6 mm drill bit; (**B**) Port result after 3 years of application. Source: Collection of the authors' pictures.

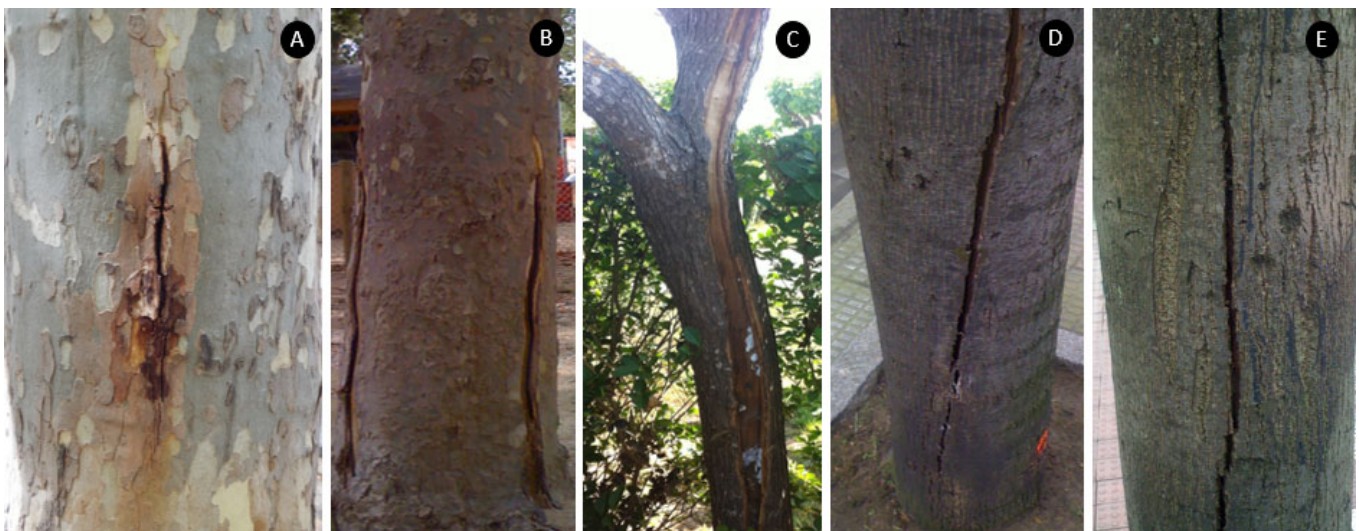

**Figure 10.** Trunk results after excess pressure years after application (**A–E**). Wounds after application using the ArborSystem® in (**A**,**B**) *Platanus hispanica*; (**C**) *Catalpa* spp; (**D**,**E**) *Tilia* spp. Source: Collection of the authors' pictures.

(3)     Nonpressurized injection methods that use a pipe or catheter attached to the trunk can expose treatments to risks in cases of accidents and vandalism. Trunks with deep, inclined holes are more susceptible to fungi, microorganisms, and rot trunks, as sap and rainwater tend to accumulate (Figure 11).

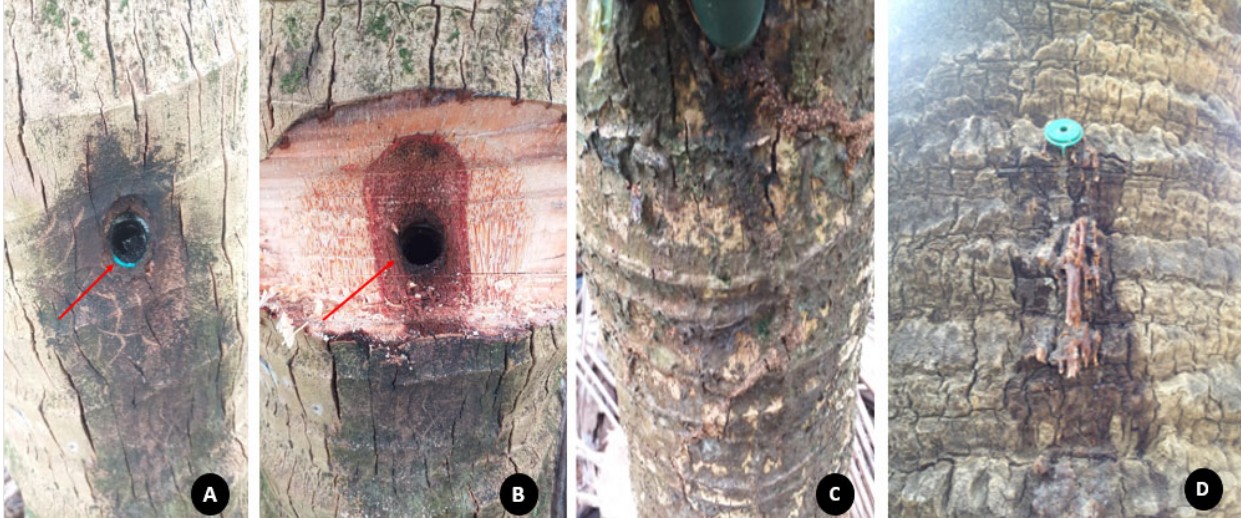

**Figure 11.** Fixation of catheter/pipe in palm stem. (**A–C**) Vita Caule® catheter/pipe break, port exposure, and damage to the coconut stem. The red arrow at A indicates that the catheter/pipe was broken inside the stem, and in B, the other palm with an area of damaged vascular bundles using this permanent accessory; (**D**) Exudation of the palm stem using SOS Palm®. Source: Collection of the authors' pictures.

(4)     Do not leave ports exposed and apply pastes or healing products to prevent the proliferation of microorganisms/pests.

(5)     Develop new formulations focused on endotherapeutic use using products that replace synthetic pesticides and antibiotics in the control of pests and diseases with natural products such as essential oils with antimicrobial/antibacterial content [153,154] and less harmful products that reach the target and/or that make it less attractive to pest attack. Since specific formulations for endotherapy are extremely limited, there is

an untapped market to be explored with new bioformulations. This may represent one of the greatest challenges to overcome in the coming years.

(6)     The extensive application of pesticides and antibiotics without criteria has been the subject of many questions regarding dosage, viscosity, and concentration of the active ingredient during applications that may create resistance in pathogens. New formulations with adjuvant action could include an application for multiple pathogens and control the entire pathosystemic problem. This lack of information prevents the determination of application intervals, treatment duration, and maximum residue limit assessments for fruit trees.

(7)     In pressurized injection methods, depressurization of the system after the plug was installed in the tree was not approached in the articles. Removing the air from the system (plug-tree) so that the applied product competes with air for space is essential to prevent cracks in the bark and an embolism that can lead to the tree's death. A simple mechanism could introduce products via endotherapy and prevent clogging and leakage during application.

(8)     The implementation of endotherapy as a trend within the NBS can contribute to the interest of more researchers for solutions inspired by efficient application techniques using less harmful products with more sustainable proposals.

## 7. Conclusions and Perspectives

Endotherapy has been made prominent by helping overcome unsustainable practices, enabling precision agriculture with some innovative technological alternatives, and considerably increasing trees' lifespans. It is believed that over the next few decades, endotherapy will create new opportunities that will contribute to the application of chemical and biological products in perennial crops.

Digital transformation and the automation revolution could quicken the pace of creating: (a) new technologies for robotic applications and artificial intelligence through sensors to detect, prevent, and treat diseases and pests, and high-efficiency endotherapy automatic equipment that replaces long hours of exhausting and repetitive physical work; (b) developing more, higher sustainable products for applications; (c) standardization of safe, sustainable methodologies and protocols to explore participatory ways of communication and collaboration between endotherapy companies, researchers and stakeholders. These demands require creative problem-solving and considerable investment in applied research in the science and technology sector. Governmental agencies and businesses should provide incentives, as the social and environmental benefits will spur the economy and boost competitiveness by forcing producers to do more with less farmland. This is not an easy task, and one can foresee a series of technical, strategic, and commercial challenges that must be faced, such as adequate pressure to introduce new products, improve dosages (concentration and viscosity), develop products designed for a particular crop, and treatment duration.

An economic evaluation of productive systems may result in developments that lead to new research areas. These may be part of integrated pest management practices. With accurate information, it will be possible to avoid excessive pesticide and fertilization use, saving time and energy, to reduce costs while obtaining greater productivity and quality, contributing to a sustainable future.

**Supplementary Materials:** The following supporting information can be downloaded at: https://www.mdpi.com/article/10.3390/agriculture13071465/s1, **Excel interactive dashboard**.

**Author Contributions:** Conceptualization, J.A.F.; methodology, J.A.F. and L.B.E.; software, J.A.F. and S.C.N.Q.; validation, L.B.E., S.C.N.Q. and C.B.G.B.; formal analysis, S.C.N.Q. and C.B.G.B.; investigation, J.A.F.; resources, L.B.E.; data curation, J.A.F.; writing—original draft preparation, J.A.F. and L.B.E.; writing—review and editing, C.B.G.B. and S.C.N.Q.; visualization, L.B.E., J.A.F., S.C.N.Q. and C.B.G.B.; supervision, S.C.N.Q. and C.B.G.B.; project administration, S.C.N.Q. and C.B.G.B.;

funding acquisition, S.C.N.Q., C.B.G.B., L.B.E. and J.A.F. All authors have read and agreed to the published version of the manuscript.

**Funding:** This research was funded by FAPESP 2017/22110-3 and 2012/18318-4. This work also was financially supported by the National Institutes of Science and Technology (INCTs) [grant numbers: FAPESP/INCT 2014/50867, CNPq 465389/2014-7]. C. B. G. Bottoli acknowledges CNPq for a research fellowship [CNPq 309363/2018-7].

**Institutional Review Board Statement:** Not applicable.

**Data Availability Statement:** Data is contained within the article or Supplementary Material.

**Acknowledgments:** The authors would like to acknowledge Paulo Dias from Empresa Digital for technical support for the development of the Excel Table and Carol H. Collins (*in memoriam*) for English assistance. Some pictures were created with BioRender.com.

**Conflicts of Interest:** The authors declare no conflict of interest.

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
