# Peer review of "Vegetative Endotherapy—Advances, Perspectives, and Challenges"

_agriculture, doi:10.3390/agriculture13071465_

Round 1

Reviewer 1 Report

The manuscript titled "Vegetative Endotherapy -Advances, perspectives, and challenges" is a review paper, focused on  the endotherapeutic techniques, showed the overview of the concepts, and discussed for existing injection methods. The contents were well-organized, comprehensive, and fully understood. This review with supplemental materials could be very useful for plant protection. There is only a few points that authors should revise in this ms. 

I feel Table 2-5 could be integrated. These lists are somewhat redundant. Also, This could be treated as supplementary material.

In Table 4, authors cited the litratures No. 140 and 141, but are they truly related to endotherapy? Selected key words used for searching may not always be appropriate for the endotherapy-related litratures. Authors should "polish" the database.

Author Response

Response to Reviewer 1 Comments

The manuscript titled "Vegetative Endotherapy -Advances, perspectives, and challenges" is a review paper, focused on  the endotherapeutic techniques, showed the overview of the concepts, and discussed for existing injection methods. The contents were well-organized, comprehensive, and fully understood. This review with supplemental materials could be very useful for plant protection. There is only a few points that authors should revise in this ms. 

Answer: We would very much like to thank you for your time in reading this manuscript. Also, we appreciate all suggestions.

I feel Table 2-5 could be integrated. These lists are somewhat redundant. Also, This could be treated as supplementary material.

Answer: We appreciate your suggestion. This was a frequent discussion among the authors, as we did not want to generate yet another supplementary material that would make the reader shift to other supplementary material. Also, we tried to condense all the table contents into a single table, and it got really confusing. And we already have all this elaborated in an illustrated and interactive way in the Excel Dashboard table. Therefore, we decided to keep the tables. 

In Table 4, authors cited the litratures No. 140 and 141, but are they truly related to endotherapy? Selected key words used for searching may not always be appropriate for the endotherapy-related litratures. Authors should "polish" the database.

Answer: We appreciate the suggestion, but we did the search and did a rigorous reading of all the works. When the paper was somehow related to endotherapy, they were included in the work. Regarding works 140 and 141, which are by the same authors, they applied a cesium radioisotope to study the translocation. These works are very important in the literature because they can be visualized through analysis and to trace the improvement in treatments. Also, the first report in the literature on endotherapy proposed by Da Vinci was also applying a radioactive element such as arsenic to apple trees, described on page 2 of this review.

Reviewer 2 Report

This is a very comprehensive review of Endotherapy and covers most aspects very well.  

Author Response

Response to Reviewer 2 Comments

General comments

This is a very comprehensive review of Endotherapy and covers most aspects very well. However, this review assumes that the technique works well but I think a section on the efficacy of treatments would add much to this paper. See my detailed comments concerning Section 5.

Response: We thank you for taking the time to read our review and your suggestions. To arrive at this version, many years passed between readings and discussions between the authors. It is important to point out that many other important pieces of information were removed because the paper was too long. We consider your suggestion very important, and it may be left for the next review to write only about the effectiveness of the treatments. However, this version comprises more of an introduction to endotherapeutic methods.

Specific comments

Line 72: (UV-rays) NOT ray

Line 114: “..its concepts, which can prevent its advancement…”

Answer: Thank you. We appreciate your suggestion, and changes have been made to the text.

LInes 119-120 partially repeat

Response: Thank you. We agreed with your suggestion and removed the sentence.

Lines 113-114. Suggestion: “..publications on endotherapy including the first technique definition framework completed in 2020 [reference], which will be updated here.”

Response: We did not publish this paper in 2020. We would just like to point out that we finalized this first version in 2020, and it is only in 2023 that we update and manage to submit it. We just would like to leave this information highlighted in the text. It's really important.

Line 205: Why does Dixon and Joly have a date while other references are numbers? And where is Dixon and Joly in the list of references?

Response: The numbers are to estimate the probable date of these researchers' experiments, which shows a very old theory. We chose to use more current references that still use this theory to explain the translocation of products by endotherapy.

Figure 2: the print is too small to read easily

Answer: This is just an illustrative image to encourage the reader to look for the interactive table in the supplemental materials.

Line 287: “Endotherapy studies were on five tree groups: arboreal, forest and urban, coniferous trees, fruit trees, and ornamental trees with 11%, 24%, 11% and 54% of studies in each of the respective categories.”

Response: We appreciate your suggestion, and changes have been made to the text.

Line 291: 3.1 Crops

Response: We appreciate your suggestion, and changes have been made to the text.

Table 2: Valencia orange trees [69] NOT oranges

Change Citrus sinensis [L.] Osbeck) to Citrus sinensis (L.) Osbeck

Answer: We appreciate your suggestion, and changes have been made to the text.

In Table 2, why have a different entry when the author is cited?

Change to Citrus paradisi Macfad [59, 61, 62]

Change to Citrus aurantifolia Swingle [49, 73, 74]

Malus domestica Borkhausen : name is not in brackets

Response: We appreciate your suggestion, and changes have been made to the text. We consider it important to specify the cultivars/variety because there is a lot of information about them in the works, and this can be important for the reader.

LItchie trees should either be Litchi or Lychee trees

Response: We appreciate your suggestion, and changes have been made to the text.

Line 306: Endotherapy is preferable to pesticide spraying as it reduces the risk to the applicator, the human population and the environment.”

Answer: We appreciate your suggestion, and changes have been made to the text.

Near end of Table 3: Prunus serotina Ehrhart

Response: We appreciate your suggestion, and changes have been made to the text.

Section 3.2

Line 366: “An estimation of dosage….”

Response: We appreciate your suggestion, and changes have been made to the text.

Line 517: “..13% did not provide information concerning the method used….”

Answer: We appreciate your suggestion, and changes have been made to the text.

  1. Analysis of endotherapeutic applications

In Lines 545+ and Lines 601-ff, the results of different methods of application methodologies are evaluated, but as mentioned in general comments, the authors should present analyses of the overall efficacy of treatments to demonstrate how well treatments worked. A section on overall efficacy could use many of the references already cited.

Response: Some works were selected for discussion and the most interesting ones between lines 545 and 601 evaluated the endotherapeutic treatment which is the focus of this review. It is very difficult to discuss treatment efficiency when working with different species and treatments. Therefore, we think the suggestion is super valid and will be for future work because this review is too long.

Line 688: “(Figures 7, 8)” omit “e” here and line 703.

Good data on examples of damage from driliing so give similar examples of where it worked,

Response: We appreciate your suggestion, and changes have been made to the text. We have some upcoming papers that will be submitted in the coming months, and then we will be able to publish some pictures and explain these results. As it involves a patent, we chose to let this be disclosed in an article that was not a review because it could be broken down in greater detail. This review is so long. At this point, we chose to highlight the precautions that must be taken to avoid errors like these ones.